# Uncertainty-Informed Active Learning using Monte Carlo Dropout for Risk Stratification in Carotid Ultrasound Imaging

Theofanis Ganitidis
School of Electrical and Computer Engineering
National Technical University of Athens
Athens, Greece
theogani@biosim.ntua.gr

Maria Athanasiou
School of Electrical and Computer Engineering
National Technical University of Athens
Athens, Greece
orcid.org/0000-0003-1575-9100

Konstantina S. Nikita
School of Electrical and Computer Engineering
National Technical University of Athens
Athens, Greece
orcid.org/0000-0001-8255-4354

*Abstract*—The integration of model uncertainty quantification in clinical decision support systems, incorporating machine learning models, can augment the models' reliability and robustness against domain shifts while also promoting user confidence and trust. In the present study, an uncertainty-informed active learning approach, leveraging Monte Carlo dropout for uncertainty estimation, is proposed towards the development of a deep learning model able to classify carotid ultrasound images as high-risk and low-risk for cardiovascular disease. An auxiliary dataset (CUBS) is employed for the initial model's development and fine-tuning as well as the optimization of the Monte Carlo dropout's hyperparameters. A dataset (87 B-mode ultrasound sequences) from ATTIKON hospital is subsequently utilized within the framework of active learning for model retraining based on the selection of the most informative samples according to the Monte Carlo dropout uncertainty estimation. In this context, the use of three active learning strategies is investigated, including uncertainty rank selection, pseudo-labeling for certain samples, and pseudo-labeling with variable sample weighting. The obtained results indicate that pseudo-labeling with variable sample weighting yields the best performance, achieving an AUC of 87.28% with only 21 annotated samples, which account for 30% of the total training data. Thus, this work provides evidence regarding the ability of uncertainty quantification and active learning to reduce labeling costs while maintaining model performance and enhancing the robustness and reliability of cardiovascular risk prediction models.

*Keywords—Deep learning, Monte Carlo dropout, Uncertainty estimation, Active learning, Cardiovascular risk prediction*

## I. INTRODUCTION

Recent advancements in deep learning (DL) have revolutionized various fields, including medical image analysis [1], where DL models have shown significant potential in automating and enhancing diagnostic processes. Particularly in the realm of cardiovascular health, deep neural networks (DNNs) have been instrumental in disease detection [2], risk stratification [3], and prognosis [4] based on imaging data. Despite their promising performance, a critical challenge toward the reliable deployment of these models in clinical settings remains unaddressed: their unpredictable behavior on data that deviates from the training distribution, known as domain shift [5]. This issue limits the generalizability of DNN models and raises concerns about their clinical utility, especially in the case of diverse patient populations and imaging conditions. Various approaches have been proposed towards addressing domain shift in machine learning, such as Source-Free Domain Adaptation (SFDA) techniques that adapt models to new data without considering source data [6], [7]. Uncertainty quantification plays a crucial role in enhancing these methods by enabling more reliable and robust adaptation to domain shifts, ultimately improving model generalization and performance in various tasks. In this context, Bayesian Neural Networks (BNNs) have been shown to improve target self-training by better estimating pseudo-label uncertainty in SFDA for semantic segmentation [8]. Furthermore, probabilistic source models with uncertainty estimation are able to contribute in identifying target data points lying outside the source manifold, thus improving adaptation robustness in SFDA scenarios [9].

In the context of cardiovascular risk assessment from carotid ultrasound images, the ability to accurately quantify model uncertainty [10] is vital. Clinicians rely on confidence estimates to make informed decisions, particularly when dealing with life-threatening conditions. Standard DL models, however, typically do not provide a measure of uncertainty, leading to overconfident predictions that can potentially misguide clinical decisions. This gap underscores the need for methods that can not only predict outcomes with high accuracy but also reliably estimate the confidence in those predictions.

Monte Carlo dropout is a widely adopted technique for uncertainty quantification in DL. It leverages the dropout mechanism, normally used only during training [11], to approximate Bayesian inference [12]. By performing multiple forward passes with dropout enabled, the model generates a distribution of predictions for each input, from which uncertainty measures such as total variance can be derived. This approach has been successfully applied in various domains, including digital pathology [13] and brain-computer interfaces [14], to provide uncertainty estimates alongside predictions, thereby enhancing the reliability of AI systems in critical applications.

The present study extends the application of Monte Carlo dropout to the domain of cardiovascular risk prediction using carotid ultrasound images. The study focuses on two main aspects: fine-tuning the dropout parameters to optimize uncertainty estimation and integrating these uncertainty estimates into an active learning framework. Active learning [15], which iteratively selects the most informative samples for annotation, has the potential to significantly reduce the amount of labeled data required for training while maintaining high model performance. By incorporating Monte Carlo dropout uncertainty estimation into the sample selection process, the aim of this study is to improve the efficiency and robustness of the active learning strategy in the context of cardiovascular disease risk stratification.

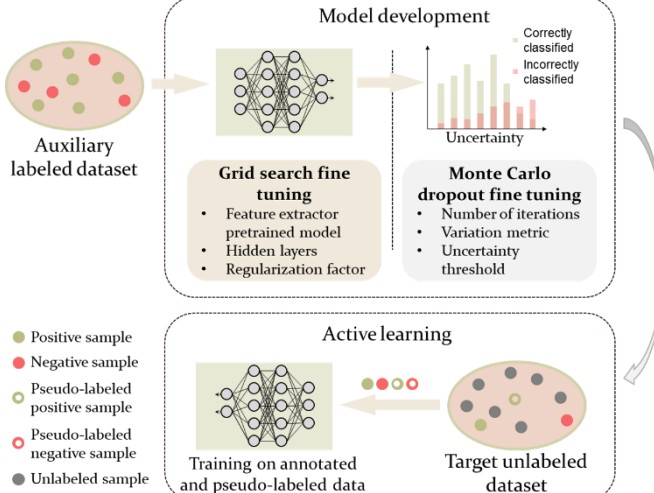

Fig. 1. Overall methodology. An auxiliary labeled dataset is used to train and validate the model's hyperparameters and fine-tune the Monte Carlo dropout configuration. In the active learning process with the target dataset, the most informative samples are selected based on model uncertainty. Predictions with uncertainties below a threshold are treated as pseudo-labels and incorporated into the training process.

A comprehensive evaluation of the proposed approach is carried out using data acquired from patients referred to Attikon General University Hospital of Athens [16], with the ultimate goal to demonstrate how fine-tuned uncertainty estimates can enhance model performance and guide the active learning process. The contributions of the present work include the identification of the optimal Monte Carlo dropout configuration, and its integration in an active learning framework featuring three different sampling strategies: uncertainty rank selection, pseudo-labeling for confident samples, and pseudo-labeling with variable sample weighting. The obtained results reveal that integrating Monte Carlo dropout uncertainty estimation into the active learning loop enhances predictive performance in cardiovascular risk stratification based on carotid ultrasound imaging by helping to reliably identify the most informative and uncertain samples. Additionally, this integration allows the model to recognize and utilize confident predictions, leading to more efficient use of labeling resources in the training process. The code related to this work is available in a public repository[1].

## II. METHODOLOGY

An overview of the proposed methodology is illustrated in Fig. 1. An auxiliary labeled dataset was firstly deployed for training and validating the risk prediction model's hyperparameters and fine-tuning the Monte Carlo dropout configuration. Different active learning strategies, involving various uncertainty-informed approaches for the selection of informative samples from a target (unlabeled) dataset, were subsequently investigated towards model retraining with the ultimate goal to boost the model's discrimination performance and enhance its reliability in the face of new data from unknown distributions.

### A. Data

The study utilized a target domain dataset [16], [17] consisting of 96 B-mode ultrasound image sequences (videos) that were acquired from 82 patients with an age range of 46 to 88 years, referred to Attikon General University Hospital of

Athens for carotid ultrasonography. The local institutional review board approved the study protocol, and all subjects gave their informed consent to the scientific use of the data. The sequences were categorized as high-risk or low-risk according to two key factors: the presence or absence of symptoms (stroke or transient ischemic attack) and the degree of carotid stenosis (narrowing of the artery lumen). Stenosis degree data was not available for all participants. The high-risk group comprised 67 sequences from patients exhibiting symptoms or presenting a stenosis degree exceeding 70%. The low-risk group consisted of 20 sequences from asymptomatic patients with a stenosis degree of 70% or less. The remaining sequences lacked information about either symptoms or stenosis degree. The final ATTIKON dataset, comprising these 87 (67 high-risk and 20 low-risk) sequences, was used within the framework of active learning for model retraining based on the selection of the most informative samples according to the Monte Carlo dropout uncertainty estimation.

The Carotid Ultrasound Boundary Study (CUBS) dataset [18] served as an auxiliary dataset, deployed for the initial model's development and the tuning of the uncertainty estimation process. CUBS is a collection of ultrasound images acquired from 1,088 patients across various healthcare centers, utilizing different ultrasound equipment. The dataset comprises 694 patients recruited from three distinct villages in Cyprus and 394 patients enrolled at the Hypertension Outpatient Clinic of the University of Pisa, Italy. For each participant, two ultrasound images were captured from both sides of the neck, resulting in a total of 2,176 images within the dataset. The Cypriot cohort within the CUBS dataset provided information regarding cardiovascular health at baseline and were subsequently monitored for up to 14 years for any new events. Data concerning both baseline and follow-up cardiovascular events was available for 689 Cypriot patients. For the purpose of this study, participants were classified as high-risk (117 individuals) or low-risk (572 individuals) depending on whether they had reported a cardiovascular event at baseline or within a range of a 3-year follow-up.

The sequences from ATTIKON dataset underwent random sampling to select a single frame, ensuring consistency between the structure of the two datasets. TABLE I. summarizes the number of data samples for each class across both datasets, highlighting the inverted class imbalance in the two datasets, with "low-risk" and "high-risk" as the minority class in ATTIKON and CUBS, respectively.

TABLE I.          CLASS DISTRIBUTION FOR EACH DATASET

|  | High-risk | Low-risk |
|---|---|---|
| **ATTIKON** | 67 | 20 |
| **CUBS** | 117 | 572 |

### B. Model Development

A DL model was developed to classify carotid ultrasound images from the CUBS dataset as high-risk or low-risk. The model's architecture consisted of a convolutional feature extractor pretrained on the ImageNet [19] dataset for feature extraction. Two different convolutional architectures were employed, the ResNet50 [20], [21] and the InceptionV3 [22]. In the realm of computer vision, these models have consistently delivered strong results [23], [24], [25], [26], [27] in CVD risk stratification tasks. In particular, ResNet50's

[1]          https://github.com/theogani/Uncertainty_based_AL_BHI

residual connections can help prevent the vanishing gradient problem, making this architecture ideal for capturing intricate details in the data, while InceptionV3's proven effectiveness in multi-scale processing enables capturing features at different levels of granularity, which is useful for diverse visual patterns. Thus, these individual strengths motivated the models' deployment towards addressing the complexity and diversity of the used dataset, with the aim of identifying subtle patterns. The outputs of the feature extraction stage comprised a total of 2048 features, which were identical for both of the employed pretrained convolutional models. The classification stage was composed of a stack of dropout and fully connected layer blocks with a rectified linear unit (ReLU) activation function, followed by a fully connected layer with a Softmax activation function and two nodes, one for each class. To prevent overfitting, early stopping based on the validation AUC score and weight decay of the fully connected layer's hyperparameters were also considered. The Adam optimizer [28] with 0.001 learning rate and the Categorical Cross-Entropy as a cost function were employed.

The model was fine-tuned through a grid search process for the identification of the optimal hyperparameters, including the number of nodes in the hidden layers and the regularization factor for the weight decay regularization. The initial dataset was split into training, validation, and test subsets using a 60-20-20 ratio. The best hyperparameters' combination was selected based on the highest AUC score obtained on the validation set.

### C. Monte Carlo dropout Uncertainty Estimation

The dropout technique is a commonly employed method for preventing neural networks from overfitting in a straightforward manner [11]. A dropout layer multiplies its input by a binary mask that is drawn according to a predefined probability distribution, randomly setting some neurons to zero in the neural network during the training phase. In contrast, during the testing phase, the output of the layer is identical to the input. As outlined in [12], the utilization of dropout at test time can be regarded as an approximation of probabilistic Bayesian inference in deep Gaussian processes, which is referred to as Monte Carlo dropout. The Monte Carlo dropout method is employed to estimate the uncertainty of the network's output in relation to its predictive distribution. This is achieved by sampling $n$ distinct dropout masks for each forward pass. Consequently, instead of a single model output, $n$ model outputs for each input sample are generated. The set of $n$ outputs can be interpreted as samples from the predictive distribution, which is useful for extracting information regarding the variability of the prediction. Quantifying the uncertainty of the model may allow for uncertain predictions to be rejected or treated differently. Fig. 2 depicts an illustration of the standard use of dropout layers and the utilization of Monte Carlo dropout for uncertainty estimation.

Within the framework of the present study, the optimization of the Monte Carlo dropout method for uncertainty estimation included fine-tuning of the number of forward passes, the uncertainty measure for estimating uncertainty, and the uncertainty threshold. The number of forward passes was varied from 10 to 150, incremented by 10, towards striking a balance between computational efficiency and uncertainty estimation accuracy. Each forward pass generated a different dropout mask, producing distinct outputs for each input sample. To quantify uncertainty from the multiple predictions, the use of the following uncertainty

measures was investigated [14]: (i) variation ratio (VR) which represents the proportion of cases not in the mode category, (ii) predictive entropy (PE) which measures uncertainty in the prediction distribution, (iii) mutual information (MI) which measures the epistemic uncertainty by capturing the model's confidence from its output, and (iv) total variance (TV) which is the sum of variances obtained for each class.

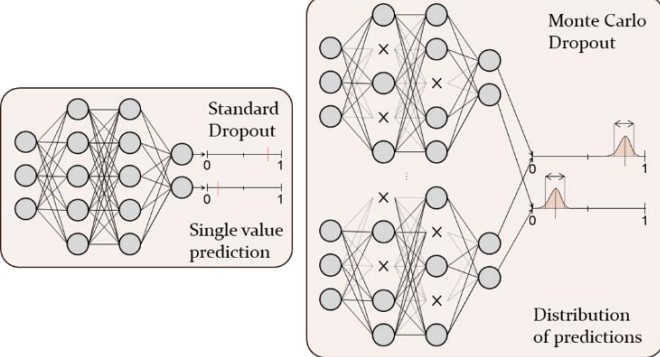

Fig. 2. Standard dropout and Monte Carlo dropout during inference mode. In standard dropout, dropout layers are not active during inference, resulting in a single prediction. In Monte Carlo dropout, dropout layers remain active during inference, generating multiple predictions and, thus, providing a distribution of outputs.

The appropriate threshold for distinguishing between certain and uncertain samples was determined experimentally, aiming at maximizing the model's ability to identify and prioritize uncertain samples for annotation in the active learning process. Different configurations were evaluated based on their ability to distinguish between correctly and incorrectly classified samples, ultimately leading to the selection of the combination which maximized the model's classification accuracy.

### D. Active Learning

The fine-tuned model, incorporating the Monte Carlo dropout, was employed in an active learning scenario, aimed at sustaining the model's performance and enhancing its reliability in classifying high-risk and low-risk carotid ultrasound images, thereby improving early detection of cardiovascular risk. The ATTIKON dataset was utilized for the development and evaluation of the uncertainty-informed active learning approach.

In this context, the model was iteratively retrained by considering new samples, selected based on their uncertainty scores. Three distinct sampling strategies were implemented during this process:

(1) *Uncertainty rank selection:* Samples were ranked by their uncertainty, and the most uncertain samples were annotated and added to the training set.

(2) *Pseudo-labeling for confident samples:* In addition to the annotation of uncertain samples, the model's predictions for samples with uncertainty below the defined threshold were used as pseudo-labels, expanding the training set without the need for additional manual annotations.

(3) *Pseudo-labeling with variable sample weighting:* Annotated samples were assigned a weight of 1 while pseudo-labeled samples were weighted at 0.5 in the loss calculation, thus balancing their influence on the model's learning process.

Given the small size of the dataset, a 5-fold cross validation scheme was applied to ensure more robust and reliable evaluation of the model's performance. In this context, the ATTIKON dataset was divided into five subsets, with the model being actively trained on four subsets and tested on the remaining one. The selected data was further split in training and validation set with a 75%-25% ratio.

## III. RESULTS AND DISCUSSION

### A. Model Development

The hyperparameters' tuning process tested 24 combinations of pretrained models, including the investigation of the use of InceptionV3 and ResNet50 as feature extractors, the hidden layer configurations [1024, 512], [1024, 256], and [1024, 512, 256], and the values of $10^{-2}$, $10^{-3}$, $10^{-4}$, and 0 for the regularization factor of weight decay. TABLE II. summarizes the top 5 combinations based on the obtained validation AUC score on the CUBS dataset.

TABLE II.  CROSS-VALIDATION METRICS OF THE TOP FIVE MODEL CONFIGURATIONS

| Pretrained model | Hidden layers | Weight decay factor | Validation AUC |
|---|---|---|---|
| InceptionV3 | [1024, 256] | $10^{-3}$ | 67.19% |
| InceptionV3 | [1024, 512] | $10^{-2}$ | 66.43% |
| InceptionV3 | [1024, 256] | $10^{-2}$ | 65.82% |
| ResNet50 | [1024, 256] | $10^{-3}$ | 64.74% |
| ResNet50 | [1024, 256] | $10^{-2}$ | 63.52% |

The best-performing configuration involved using the InceptionV3 model, followed by two hidden layers with 1024 and 256 nodes, respectively, and a weight decay regularization factor of $10^{-3}$. This configuration was evaluated on the separate test set achieving accuracy, sensitivity, balanced accuracy, and AUC scores of 62.3%, 65.2%, 63.4%, and 68.44%, respectively.

### B. Uncertainty Estimation Fine-Tuning

The fine-tuning of the hyperparameters of the Monte Carlo dropout method focused on optimizing the number of forward passes $n$ and selecting the most appropriate uncertainty measure among the variation ratio, the predictive entropy, the mutual information, and the total variance for uncertainty estimation. Discrimination performance metrics were applied to assess the model's ability to distinguish between correctly and incorrectly classified samples for different combinations of $n$ forward passes and uncertainty measures. Fig. 3-5 illustrate the obtained AUC, accuracy and Youden's index for these combinations on the CUBS dataset. The obtained results indicate that the combination of 130 forward passes and total variance as uncertainty measure, was consistently the most effective one. This combination led to the maximization of all evaluation metrics of the model's discrimination performance.

Fig. 6(a)-(d) further highlights the superiority of the identified optimal combination of 130 forward passes and total variance by depicting the distributions of the uncertainty estimation for the correctly and incorrectly classified samples for each uncertainty measure and the corresponding optimal number of forward passes based on the obtained AUC score. As shown there, total variance displayed a broader range of uncertainty values compared to variation ratio, mutual information, and predictive entropy, which exhibited a narrower range, predominantly consisting of high uncertainty values. This broader range enabled better differentiation

between low, medium, and high uncertainty samples, ultimately leading to better performance. The increased effectiveness of total variance, which is the sum of variances obtained for each class, stems from its comprehensive approach to capturing the variability across all classes, making it a more robust measure of uncertainty compared to the other methods, which focus on specific aspects of uncertainty.

The selected uncertainty threshold for the optimal configuration of total variance and 130 forward passes, as determined by the Youden's index, was 0.571. This threshold achieved an optimal balance between sensitivity and specificity, resulting in the effective discrimination between certain and uncertain predictions.

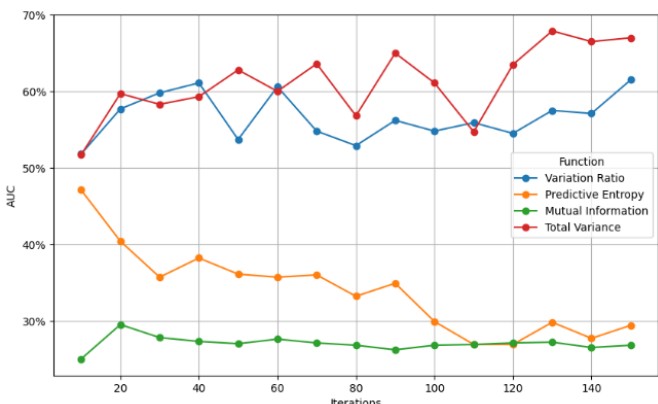

Fig. 3. Obtained AUC scores for the considered uncertainty measures across different numbers of forward passes.

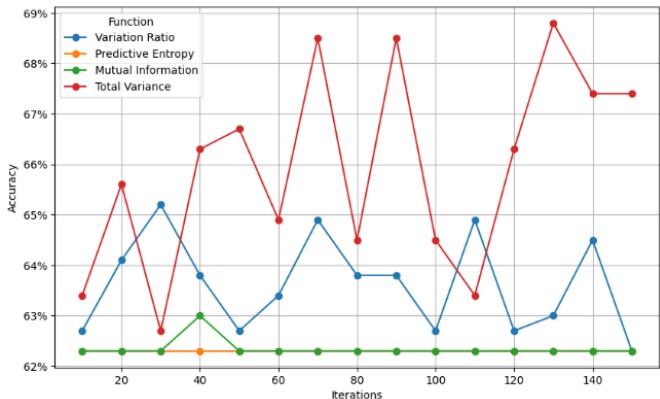

Fig. 4. Obtained accuracy for the considered uncertainty measures across different numbers of forward passes.

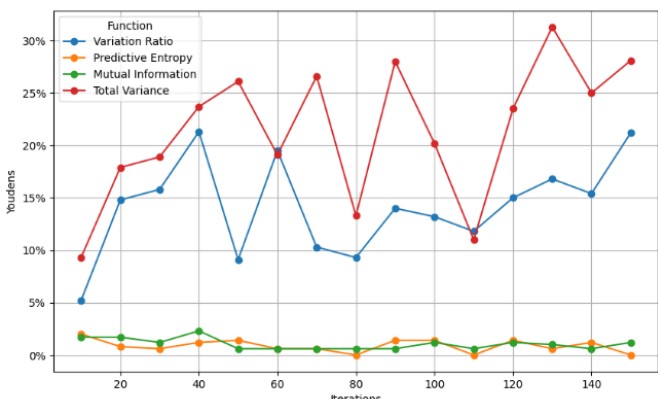

Fig. 5. Obtained Youden's index for the considered uncertainty measures across different numbers of forward passes.

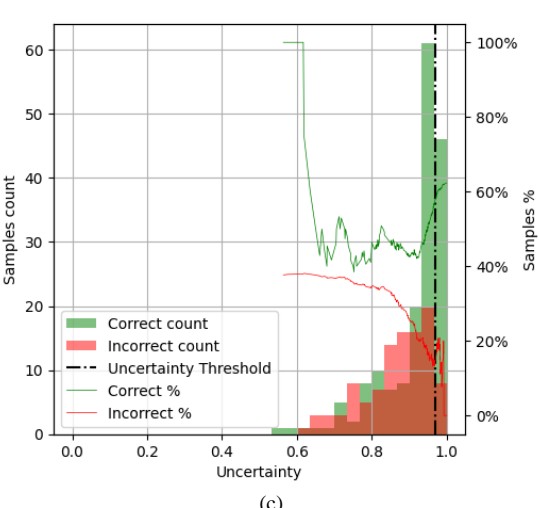

(a)

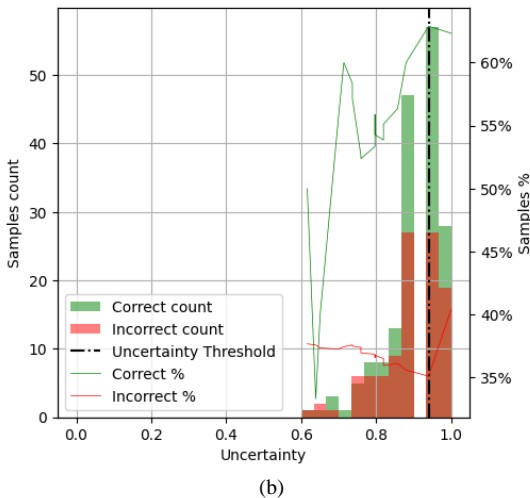

(b)

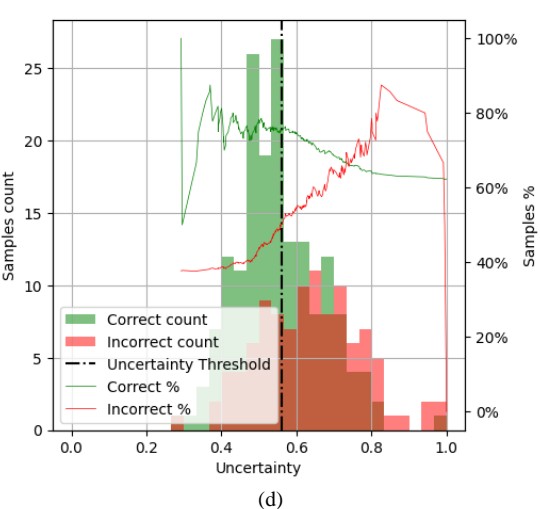

(c)

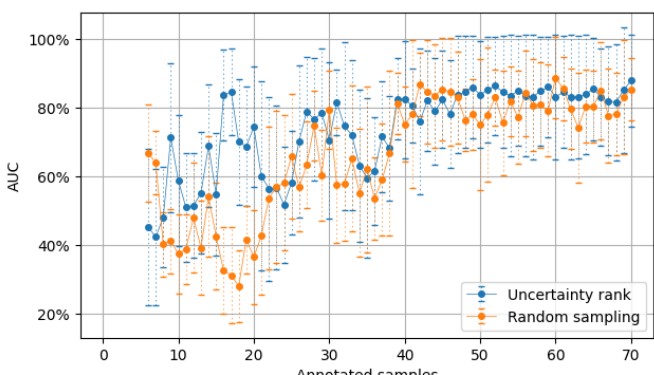

(d)

Fig. 6. Distribution of uncertainty estimation for correctly and incorrectly classified samples in the case of (a) variation ratio with 150 forward passes, (b) predictive entropy with 10 forward passes, (c) mutual information with 20 forward passes, (d) total variance with 130 forward passes. The green and red bars represent the number of correctly and incorrectly classified samples, respectively, corresponding to each uncertainty estimation value. The dash-dot line indicates the optimal uncertainty threshold based on Youden's index. The green and red lines show the percentage of correctly and incorrectly classified samples, respectively, with uncertainty values below or above the corresponding value on the x-axis.

## C. Active Learning

Fig. 7-9 illustrate the evolution of the model's discrimination performance in terms of the AUC (mean and standard deviation across the 5-fold evaluation scheme) across the iterations of the active learning process for the three different sampling strategies. The model's performance showed progressive improvement as more samples were actively selected and annotated. Notably, each of the three sampling strategies achieved performance levels comparable to the performance of the fully supervised case (100% of labeled samples) more quickly than random sampling. Specifically, the uncertainty rank selection approach, which focused on selecting the most uncertain samples for annotation, led to non-gradual improvements in model performance with each iteration. The use of 16 samples achieved performance (AUC=83.62%) equivalent to that of the fully supervised case (AUC=87.82%). However, the model's performance subsequently dropped significantly before reaching its highest value.

Fig. 7. Evolution of model performance in terms of the AUC using the uncertainty rank selection approach compared to random sampling.

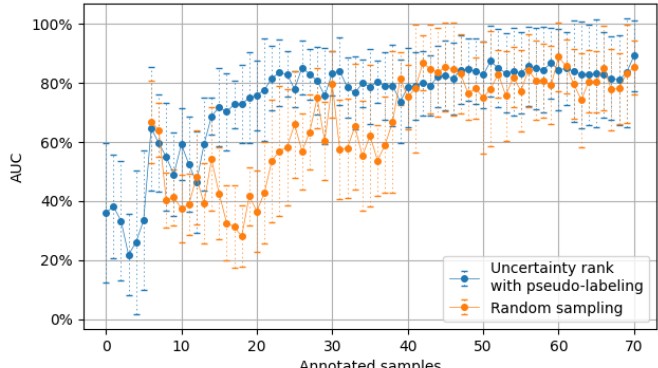

Fig. 8. Evolution of model performance in terms of the AUC using the uncertainty rank selection combined with pseudo-labeling approach compared to random sampling.

Pseudo-labeling of certain samples by using the model's confident predictions as pseudo-labels effectively expanded the training set and demonstrated a slower, more gradual improvement compared to the uncertainty rank selection alone. The model required 22 annotated samples to achieve an AUC of 81.14% but displayed a steady improvement, which indicated that the incorrectly pseudo-labeled samples caused a decrease in the obtained performance. Nevertheless, the model's consistent improvement demonstrated the ability of the adopted active learning strategy to increase the reliability of the risk prediction model.

It is noteworthy that the assignment of variable sample weights to annotated and pseudo-labeled samples during training proved to be the most effective strategy, yielding the best overall performance (AUC=87.28% ± 9.66%) with the use of 21 annotated samples, while demonstrating a steady improvement throughout the iterations, too. These results indicated the ability of variable sample weighting to balance the influence of confident predictions and actively annotated samples.

Overall, each of the three active learning strategies contributed to enhanced model performance and reliability and demonstrated their ability to address the inverted class imbalance with respect to the CUBS dataset, which was present in the ATTIKON dataset. Particularly, the variable sample weighting strategy demonstrated substantial gains, which highlighted its efficacy in active learning scenarios.

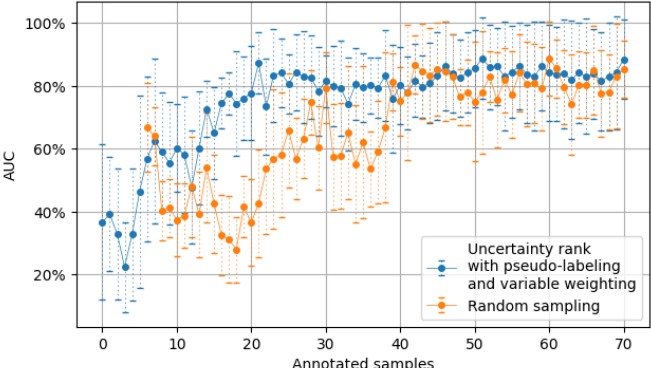

Fig. 9. Evolution of model performance in terms of the AUC using the uncertainty rank selection combined with variable sample weighting compared to random sampling.

## IV. CONCLUSION

In this study, a robust deep learning model for classifying carotid ultrasound images as high-risk and low-risk for cardiovascular disease was developed. The proposed approach utilized Monte Carlo dropout for uncertainty estimation and implemented an active learning framework to enhance model performance while minimizing labeling costs.

The obtained results underscored the importance of uncertainty estimation in enhancing the reliability and performance of deep learning models for cardiovascular risk prediction. Monte Carlo dropout provided a practical means of estimating model uncertainty, which was critical for the active learning process. By effectively identifying and prioritizing uncertain samples for annotation, significant improvements in model performance were observed. The comparison of different active learning strategies revealed that variable sample weighting not only improved the AUC but also maintained a steady performance improvement across iterations, thus demonstrating its potential for real-world clinical applications where labeling resources are limited.

Potential limitations of the present study include the use of a relatively limited target dataset, which may restrict the generalizability of the results. Moreover, the proposed approach was specifically tailored to the use case of cardiovascular disease risk stratification based on B-mode ultrasound images; further validation of the methodology on different datasets and use cases would be required to investigate its applicability to other medical imaging tasks. The absence of a stopping criterion in the active learning process also poses a challenge for the practical implementation of the approach in real-world scenarios, as the actual top performance or plateau of reaching the top performance is not determined.

Future research will focus on several key areas to further enhance the utility and applicability of the presented approach. Expanding the dataset to include a more diverse set of images from different institutions could improve the model's generalizability and robustness to domain shifts. Moreover, exploring other uncertainty estimation techniques, such as test-time augmentation [29], or similarity metrics [30] including structural similarity index, could provide alternative or complementary insights into model uncertainty. The investigation of the impact of different active learning strategies on various medical imaging tasks could help generalize the obtained findings to other domains within healthcare. The development and integration of a formal stopping criterion within the active learning framework also constitutes a critical area of future work, aiming at optimizing the number of iterations required and enhancing the model's applicability in real-world clinical settings. Other promising avenues for future work include the integration of semi-supervised learning techniques to further leverage unlabeled data as well as explainability techniques to address potential reliability and trustworthiness considerations associated with the deployment of AI-driven diagnostic tools in healthcare environments.

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
