# OpenReview forum: "Uncertainty-Informed Active Learning using Monte Carlo Dropout for Risk Stratification in Carotid Ultrasound Imaging"
_IEEE.org/EMBS/BHI/2024/Conference — IEEE BHI'24_

### Official Review · Reviewer_BYAJ · 2024-08-06
**Uncertainty-Informed Active Learning using Monte Carlo Dropout for Risk Stratification in Carotid Ultrasound Imaging**

**Overall Rating:** 7
**Confidence:** 5

**Other Quality Metrics:**

(a) Great
(b) Good
(c) Good
(d) Great

**Questions For The Authors:**

No questions.

**Strengths:**

The paper topic falls within the scope of BHI'24. The paper is well-organized and written and the use of English is good.

**Summary Of The Paper:**

In this paper, the authors employ an uncertainty-informed active learning approach to develop a deep learning model for the classification of carotid ultrasound images into high-risk or low-risk regarding cardiovascular disease. For the estimation of the uncertainty, they make use of a Monte-Carlo approach.

**Weaknesses:**

A short discussion of the limitations of the authors' approach would be useful.

---

### Official Review · Reviewer_VrRc · 2024-08-08

**Overall Rating:** 7
**Confidence:** 5

**Other Quality Metrics:**

● Clarity of writing: good
● Clinical Significance: great
● Methodological Novelty: great
● Experiments and Results: good

**Questions For The Authors:**

1.	How many samples are with high risks, and how many are with low risks in the dataset? It would be helpful to understand if the dataset is imbalanced. It would be interesting to see if the proposed method (like ensemble) can also address the imbalance issue.
2.	Since total variance is identified as the most effective uncertainty measure and the optimal number of forward passes was 130, is it necessary to present other scenarios in Fig. 6? Fig. 6a-c seem distracting. I think adding more explanation of Fig. 3-5 would be helpful if you hope to explain the results of other measures.
3.	In Fig. 7-9, for each data point of AUC, there seems to be a standard deviation range. Would you explain what it is?
4.	It is unclear how ATTIKON dataset was split. What is the AUC shown in Fig. 7-9? Is it validation or test set AUC? Would cross-validation be needed to have a solid conclusion - how many annotated samples are needed? If only one-fold was used and validation AUC was evaluated, the results and conclusions such as best performance achieved with 21 annotated samples seems a little bit absolute without considering the randomization of the training/validation samples. Comparing test set AUC seems to be more valid.
5.	In this scenario, the top performance of the model is known through using all annotated dataset and all data was utilized to determine the top performance or plateau of reaching the top performance. It would be helpful to explain or discuss how this labeling approach can be applied to a new dataset to answer questions, such as when should we say the best performance is reached and we should stop annotating?

**Strengths:**

The methodology is novel. A lot of clinical questions require risk estimation or classification. The uncertainty can be caused by development of disease, individual differences, and manual annotation when there are no objective measures. I think applying a method like this would be helpful.

**Summary Of The Paper:**

This paper presented a novel method of Monte Carlo dropout and active learning strategy to fine tune a pre-trained model with a smaller number of annotated samples in the application of risk classification.

**Weaknesses:**

The result section needs to be clearer.

---

### Official Review · Reviewer_fB3h · 2024-08-10
**Weak accept**

**Overall Rating:** 6
**Confidence:** 2

**Other Quality Metrics:**

Clarity of writing: Excellent
Clinical significance: great
Methodological novelty: fair
Experiments and results: good

**Questions For The Authors:**

Revision suggestions/comments
=============================
Methodology
-----------
- "Two different convolutional architectures were employed: the ResNet50 [20], [21] and the InceptionV3 [22]". I would like to know why this choice was made. Even if this was an intuitive decision, I think it would make sense to explain why the authors selected these two neural network models.
- Providing the analysis code in an online repository where it could be reused for further research would be good.

Discussion
----------
- The size of the ATTIKON dataset should be explicitly identified as a limitation.

Editing suggestions/comments
============================
N/A

**Strengths:**

Indeed, developing ML algorithms which could guide clinicians in terms of cardiovascular risk assessment could potentially have a huge clinical impact.

**Summary Of The Paper:**

The paper proposes the use of a specific Uncertainty quantification approach (Monte Carlo dropout) to improve Carotid Ultrasound image processing for cardiovascular risk assessment. The subject of the paper is of interest and well-written. My comments should be received only as minor improvement suggestions.

**Weaknesses:**

Methodology
-----------
- Providing the analysis code in an online repository where it could be reused for further research would be good.

Discussion
----------
- The size of the ATTIKON dataset should be explicitly identified as a limitation.

---

### Decision · Program_Chairs · 2024-09-23

Accept